# Quality of Life for Polish Women with Ovarian Cancer during First-Line Chemotherapy

**DOI:** 10.3390/healthcare11182596

**Published:** 2023-09-21

**Authors:** Grażyna Bączyk, Anna Pleszewa, Dorota Formanowicz, Katarzyna A. Kozłowska

**Affiliations:** 1Department of Nursing Practices, Poznan University of Medical Sciences, 61-701 Poznan, Poland; kkozlowska@ump.edu.pl; 2Wielkopolska Oncology Centre Poznan, 61-866 Poznan, Poland; anna.pleszewa@wco.pl; 3Department of Medical Chemistry and Laboratory Medicine, Poznan University of Medical Sciences, 61-701 Poznan, Poland; doforman@ump.edu.pl; 4Department of Stem Cells and Regenerative Medicine, Institute of Natural Fibres and Medicinal Plants-National Research, 62-064 Plewiska, Poland

**Keywords:** ovarian cancer, quality of life, QLQ-C30 and QLQ-OV28 questionnaire, first-line chemotherapy

## Abstract

Ovarian cancer is the worst prognostic gynaecological cancer and represents a grave clinical and social problem. Therefore, the study aimed to assess female patients’ emotional, cognitive, physical, and social quality of life. The study included 100 patients diagnosed with ovarian cancer and treated with chemotherapy in a day hospital setting at the Department of Radiotherapy and Gynaecological Oncology at the Wielkopolska Oncology Centre in Poznań. The patients were given a standard treatment regimen: paclitaxel 175 mg/m^2^ in a 3 h infusion and carboplatin at an AUC of 6 (5–7) following Calvert as a 1 h infusion for six cycles administered every 21 days. In addition, standardised questionnaires of the Polish version of the EORTC QLQ-C30 and QLQOV28 were used. The analysis of the collected material shows that the patients reported the highest level of general health and quality of life at the study’s first stage, i.e., before chemotherapy (mean value of 59.67 points). In contrast, the patients’ lowest level of general health and quality of life was observed in the fourth stage of the study (mean value of 45.04 points). The problem of side effects, such as nausea and vomiting, affected the entire study group and was more troublesome in the final stage of treatment for all patients. In the study’s first stage, the mean score on the nausea and vomiting symptom scale was 16 points; in the fourth stage, the mean score was 40.07. Of the clinical factors, the symptom of fatigue was the most severe health problem for the subjects. The mean score of the fatigue scale in the study’s first stage was 37.11 points, while a score of 70.33 was obtained in the fourth stage of the research. The multivariate linear regression model showed that the lack of professional activity lowers quality of life, especially combined with other side effects of chemotherapy, including hair loss in Stage IV of the study. This study shows that women with ovarian cancer undergoing chemotherapy need exceptional support from psychologists, nurses, dieticians, and physiotherapists.

## 1. Introduction

Clinical research of ovarian cancer has evolved significantly in recent years due to the changing perception of ovarian cancer as a single disease to one encompassing several different histotypes, which differ in aetiological origin, risk factors, molecular profiles, therapeutic approaches, and clinical outcomes [1]. Despite significant progress in understanding the aetiological heterogeneity of ovarian cancer and clinical advances, ovarian cancer ranks seventh among malignancies in women. It is ranked eighth as a cause of cancer deaths in women worldwide. It is currently the most significant obstacle to achieving desirable life expectancy in most countries [1,2]. Although ovarian cancer occurs less frequently than breast cancer, it is three times more lethal. Epidemiological projections indicate that ovarian cancer mortality will increase significantly by 2040. From recent data, in 2018, there were 300,000 new diagnoses and 185,000 deaths worldwide [1,2,3].

Women with ovarian cancer perceive their situation in a specific way, which mirrors differentiated behaviour. The illness triggers many life changes and presents patients with complex tasks, which they have to deal with, in many instances, for the first time [4]. The clinical manifestations of the disease significantly affect the extent to which women can functon in everyday life and even force them to give up on their previous social roles, reducing their quality of life.

Primary treatments for oncological diseases include surgery, radiotherapy, and chemotherapy [4,5]. Chemotherapy is the second primary treatment option for ovarian cancer, following surgery, and can be used in almost all stages of the disease. Women who receive chemotherapy, often with multi-drug regimens, experience many side effects. This type of treatment negatively affects health-related quality of life. Chemotherapy is essential in treating ovarian cancer, with platinum derivatives (cisplatin and carboplatin) and taxanes (paclitaxel) showing the highest activity combined with high toxicity. The side effects associated with chemotherapy adversely affect patients’ quality of life, limit the dose of cytostatic agents, and shorten the duration of treatment [4,5]. Chemotherapy for ovarian cancer causes acute adverse effects, including nausea and vomiting, loss of appetite, and gastrointestinal mucosal reactions. The incidence of nausea and vomiting in cancer patients is approximately 40–70% [6,7]. For these reasons, supportive treatment is becoming as important as chemotherapy itself, as it aims to alleviate these symptoms [6,7].

If unchecked, this type of adverse treatment effect causes significant problems in terms of mental functioning, resulting in severe disorder in patients’ daily life activities and negatively affecting their quality of life [6,7]. 

Therefore, therapeutic teams accompany the patient in the treatment process. Quality of life (QoL) studies should be a source of information on the patient’s assessment of their ongoing life situation while undergoing therapy. These studies provide valuable information on how to improve the patient’s treatment, allowing comparison of the potential benefits of the proposed treatment for the patient. Many authors see the primary goal of quality of life research as such, focusing on the unfavourable consequences and establishing the relationship between the expectations and aspirations of the patients and their actual experiences. All the components mentioned above, based on the confrontation with the disease and its treatment, make quality of life research possible [8,9,10,11]. 

There is an increasing emphasis in clinical trials on patient quality of life during and after treatment. Particular attention is placed on parameters such as progression-free survival, overall survival, objective response rate, duration of treatment response, and clinical benefit rate. Therefore, quality of life data during and after therapy are paramount. To collect these data, patients use apps to help generate reports on quality of life, functionality, health status, and overall health. In oncological diseases, survival is not the only goal of treatment; quality of life often plays an essential role in treatment [9,12].

The main aim of this study is to analyse the quality of life of patients treated for ovarian cancer during first-line chemotherapy in a single-day hospitalisation using specific scales: the Cancer Quality of Life Questionnaire (QLQ-C30) and Ovarian Cancer Quality of Life Questionnaire (QLQ OV28). 

### 1.1. Particular Objectives

The study’s objective is to assess the quality of life in terms of the emotional, cognitive, physical, and social aspects of patients treated for ovarian cancer during first-line chemotherapy using four measurements.Four measurements, including the chemotherapy series (study stage), are compared to patients’ quality of life.Finally, the relationship between the selected parameters and patients’ quality of life concerning the chemotherapy series (study stage) is studied.

### 1.2. Hypotheses

Chemotherapy reduces the quality of life of respondents.Considering the severity of the side effects of chemotherapy, the subjects’ quality of life decreases with subsequent courses of drug administration.

### 1.3. Tests

The study included 100 patients with a diagnosis of ovarian cancer hospitalised in the Department of Radiotherapy and Gynaecological Oncology at the Wielkopolska Oncology Centre in Poznań. These patients qualified for surgery in accordance with the Centre’s Interdisciplinary Committee and then for treatment with first-line chemotherapy in a one-day regimen. After surgery, the patients received standard chemotherapy according to the treatment regimen. Disease staging was analysed in the first stage of the study according to the 2014 International Federation of Gynecology and Obstetrics (FIGO) [13]. The most crucial criterion for participation in the study was patient consent. 

The following inclusion criteria were adopted for the study:(a)Post-operative patients scheduled for first-line chemotherapy treatment;(b)Histopathologically confirmed ovarian cancer;(c)No other cancer and no COVID-19;(d)Good general condition (0–1) according to the Zubrod-ECOG-WHO scale (https://www.mp.pl/interna/table/016_8031, accessed on 1 March 2019) and performance scale according to the Eastern Cooperative Oncology Group Zubroda-ECOG-WHO scale, a tool that allows determination of general condition (0—typical performance: ability to perform everyday activities without limitations; 1—the presence of disease symptoms, ability to walk and do light work);(e)white race, Polish nationality.

The following exclusion criteria were adopted for the study:(a)Previous surgery of the reproductive organs and chemical treatment;(b)Another cancer;(c)Palliative treatment.

The study ran from 1 May 2019 to 30 October 2020.

The first-line chemotherapy treatment regimen included administration of paclitaxel at a dose of 175 mg/m^2^ in a 3 h infusion and carboplatin at an AUC of 6 (5–7) following Calvert as a 1 h infusion. The chemotherapy comprised six cycles administered every 21 days. In Figure 1, the course of the study is presented.

## 2. Methodology

Polish versions of the scales were used to assess the quality of life: QLQ-C30 and QLQ OV28. 

### 2.1. Standardised Questionnaires

#### 2.1.1. EORTC QLQ-C30 Version 3.0

The European Organisation for the Research and Treatment of Cancer Quality of Life Questionnaire-Core 30 (EORTC QLQ-C30) was used in this study. The Quality of Life Study Group created the EORTC QLQ-C30 questionnaire in accordance with the European Organisation for the Research and Treatment of Cancer (EORTC). It is a fundamental tool for measuring the quality of life in the cancer patient population; it does not consider the cancer’s form, type, or location. It consists of 30 questions using a 4-degree Likert scale. The exceptions are two questions on health status and general quality of life, which use a 7-point scale. The measurement of the patient’s functioning includes physical and emotional functioning, functioning in social roles, cognitive and social functioning, and overall quality of life. Assessment of the impact of symptoms on quality of life includes fatigue, nausea and vomiting, dyspnoea, pain, sleep disturbances, constipation, diarrhoea, and loss of appetite. In addition, it is possible to assess the impact of the disease on the patient’s financial situation. The QLQ-C30 questionnaire assesses the quality of life in 15 dimensions. In each of these, the quality of life is expressed on a 0–100 scale. The first 6 of these are functional scales, in which a higher score indicates a higher level of functioning, and a higher level of general health means a higher quality of life. The remainder are symptom scales, in which a higher score indicates an intense severity of disease symptoms and, therefore, a lower quality of life. 

EORTC approval for the questionnaire was emailed via the European Organisation for Research and Treatment website on 12 February 2019.

#### 2.1.2. EORTC QLQ-OV28

The European Organization for the Research and Treatment of Cancer Quality of Life Questionnaire—Ovarian Cancer Module (EORTC QLQ-OV28) is an ovarian module of the questionnaire, containing questions on physical and psychological symptoms in ovarian cancer patients, taking into account disease state and treatment modality (i.e., surgery, chemotherapy, radiotherapy). The ovarian module should always be completed together with the QLQ-C30 questionnaire. 

The questionnaire contains 28 questions grouped into three functional scales assessing self-perception, sexuality, attitude to illness, and treatment. Assessment of the impact of symptoms on quality of life includes gastrointestinal disorders, peripheral neuropathies, menopausal symptoms, and other effects of chemotherapy, as well as the impact of hair loss on quality of life. It consists of 28 questions assessed using a 4-degree Likert scale. The quality of life assessment ranges for each scale from 0–100. For all symptoms and functional scales on the QLQ-OV28, a higher score indicates a higher severity of the problem and, therefore, lower quality of life.

EORTC approval was obtained for using the scales mentioned above, and the key was received by email via the European Organisation for Research and Treatment website on 12 February 2019.

### 2.2. Documentation of the Patient’s Medical History

Laboratory results were analysed from each patient’s medical history, including haemoglobin, erythrocyte, leukocyte, neurocyte, platelet, and CA125 antigen levels. In addition, comorbidities and stages of disease according to the 2014 FIGO [10] were analysed in the study’s first phase. A subsequent article will include laboratory results regarding quality of life analysis.

### 2.3. Questionnaire

A self-administered survey questionnaire comprised five questions concerning age, place of residence, marital status, education, and professional activity.

### 2.4. Ethics

The study was conducted in accordance with the Declaration of Helsinki and was approved by the Ethics Committee of the Poznan University of Medical Sciences and registered under reference numbers 46/16 and 564/16 (all patients gave informed consent, and the Ethics Committee of Poznan University of Medical Sciences approved the study registered as case Nos. 46/16 and 564/16). Participation in the survey was voluntary and anonymous, and all participants in the study gave their informed consent to participate. The informed consent form contained information about the study, its purpose, the method of responding to questions, and the possibility of withdrawing from the study at any time without incurring any consequences.

### 2.5. Statistical Methods

Analysis of the quantitative variables (i.e., expressed in numbers) was carried out by calculating the arithmetic mean, standard deviation, median, and quartiles.

Analysis of the qualitative (i.e., non-numeric) variables was carried out by calculating each value’s number and percentage of occurrence.

Qualitative variables were compared across groups using the chi-square test (with Yates correction for 2 × 2 tables) or Fisher’s exact test where low expected counts appeared.

Comparisons of quantitative variables across three or more groups were made using the Kruskal–Wallis test. When statistically significant differences were detected, a post hoc analysis was performed with Dunn’s test to identify statistically significantly different groups.

Comparisons of quantitative variables across the four repeated measures were made using the Friedman test. Once statistically significant differences were detected, a post hoc analysis (Wilcoxon paired *t*-tests with Bonferroni correction) was performed to identify statistically significant measurement differences.

Multivariate analysis was used to determine the factors influencing the quality of life and to assess their significance and the amount of variance explained by these factors.

The variables of age, marital status, education, professional activity, comorbidities, and FIGO scale were introduced into the model.

Multivariate analysis was used for the QLQ-OV 28 scale for Stages I and IV. 

Statistical analysis was performed in R. version 4.1.0 using the R Core Team (2021) method (R: a language and environment for statistical computing, R Foundation for Statistical Computing, Vienna, Austria. URL https://www.R-project.org/) (accessed on 30 December 2019).

## 3. Results

### 3.1. Characteristics of Respondents

One hundred women participated in Stage I of the study. In comparison, 94 respondents participated in Stage IV. Five women did not continue chemotherapy due to the development of peripheral polyneuropathy: three women aged 61 years and over and two women aged 41–50. Moreover, one woman aged 41–50 continued treatment in a hospital closer to home after the first administration.

Almost 38% of the respondents had completed the secondary education level. More than half of the women in the study (64%) were pensioners, and 36% worked professionally. In 68% of the patients, the stage of ovarian cancer was defined as Stage III according to the FIGO classification, with 63% in Stage IV of the study (Table 1).

### 3.2. Analysis of the Quality of Life as Based on the QLQ-C30 and QLQ-OV28 Scales of the Subjects in the Different Stages of the Study (Table 2)

The analysis showed statistically significant differences between the different stages of the study for all quality of life scales. 

The mean values for general health, physical functioning, role functioning, emotional functioning, cognitive functioning, and social functioning were higher in the first stage of the study compared to the mean values in the subsequent stages of the study. In the fourth stage of the study, the mean values were the lowest, indicating the lowest quality of life of the subjects.

Statistically significant differences were found for all areas of the scale across the stages of the study (*p* < 0.001) (Table 2).
healthcare-11-02596-t002_Table 2Table 2Comparison of patients’ quality of life by stage of the study based on QLQ-C30 functional scales.Quality of LifeStage IStage IIStage IIIStage IV*p*General health/QOL Mean ± SD59.67 ± 16.3153.54 ± 14.3948.65 ± 16.5345.04 ± 18.59*p* < 0.001 *median58.33505041.67
quartiles50–66.6750–66.6737.5–58.3333.33–58.33I > II > III, IVPhysical functioning Mean ± SD74.27 ± 20.1965.45 ± 19.6251.11 ± 22.3845.46 ± 21.99*p* < 0.001 *median8066.6753.3346.67
quartiles60–93.3346.67–8036.66–66.6726.67–65I > II > III > IVFunctioning in rolesMean ± SD69 ± 24.2858.08 ± 23.0146.13 ± 24.3839.72 ± 26.22*p* < 0.001 *median66.6766.675033.33
quartiles50–10033.33–66.6733.33–66.6716.67–66.67I > II > III.IVEmotional functioningMean ± SD55.75 ± 18.9447.47 ± 18.6537.12 ± 19.0628.28 ± 22.14*p* < 0.001 *median58.335033.3333.33
quartiles41.67–66.6733.33–66.6725–508.33–47.92I > II > III > IVCognitive functioning Mean ± SD77 ± 19.7971.38 ± 21.5763.97 ± 21.5257.98 ± 22.36*p* < 0.001 *median83.3366.6766.6766.67
quartiles66.67–10050–83.3350–83.3350–66.67I > II > III > IVSocial functioning Mean ± SD63.5 ± 21.5451.68 ± 22.2739.39 ± 23.0235.11 ± 26.04*p* < 0.001 *median66.675033.3333.33
quartiles50–66.6733.33–66.6725–66.6716.67–66.67I > II > III, IV*p*—Friedman test + post hoc analysis (Wilcoxon matched pair tests with Bonferroni correction). * statistically significant relationship (*p* < 0.05). QLQ-C30: functional scales—higher value indicates a higher level of functioning/quality of life; range 0–100.

The mean values in the functional and symptom scales increased with the order of the study stage, indicating higher problem severity and lower quality of life for the patients. Statistically significant differences were found for all areas of the symptom scales in the different stages of the study (*p* < 0.001). There was the highest increase in mean values for “hair loss”, with a mean value of 8.67 in Stage I, while in Stage IV of the study, the value increased to 65.96 (Table 3).

### 3.3. Women’s Quality of Life According to the QLQ-OV28 Functional and Symptom Scales at Stage I of the Study

Perception of One’s Own Body Domain—See Appendix A.Sexuality Domain—See Appendix A.Approach to Disease/Treatment Domain—See Appendix A.Gastrointestinal Symptoms Domain—See Appendix A.Peripheral Neuropathy Domain—See Appendix A.Hormonal/Menopausal Symptoms Domain—See Appendix A.Other Side Effects of Chemotherapy Domain—See Appendix A.Hair Loss Domain—See Appendix A.

### 3.4. Women’s Quality of Life According to the QLQ-OV28 Functional and Symptom Scales at Stage IV of the Study

#### 3.4.1. Perception of One’s Own Body Domain

Table 4 presents the results of a multivariate analysis for the QLQ-OV28 scale in the perception of one’s own body domain.

The multivariate linear regression model showed that patients aged 61 and over had a perception of one’s own body domain score that was, on average, 38.395 points lower than those aged up to 50 years. A lack of professional activity was associated with an increase in the score for the perception of one’s own body domain by an average of 34.2 points. 

The R^2^ coefficient for this model was 13.62%, which means that 13.62% of the variability of the perception of one’s own body domain result was explained by the variables included in the model (see Table 4).

#### 3.4.2. Sexuality Domain

Table 5 presents the results of a multivariate analysis for the QLQ-OV28 scale in the sexuality domain. The multivariate linear regression model showed that none of the analysed features was a significant independent predictor of the sexuality domain score (all *p* > 0.05).

The R^2^ coefficient for this model was 15.71%, which means that 15.71% of the variability of the sexuality domain result was explained by the variables included in the model (see Table 5).

#### 3.4.3. Approach to Disease/Treatment Domain

Table 6 presents the results of a multivariate analysis for the QLQ-OV28 scale in the approach to disease/treatment domain. 

The multivariate linear regression model showed that none of the analysed features was a significant independent predictor of the disease/treatment domain score (all *p* > 0.05).

The R^2^ coefficient for this model was 11.19%, which means that 11.19% of the variability of the disease/treatment domain result was explained by the variables included in the model (see Table 6).

#### 3.4.4. Gastrointestinal Symptoms Domain

Table 7 presents the results of a multivariate analysis for the QLQ-OV28 scale in the gastrointestinal symptoms domain.

The multivariate linear regression model showed the following:-The status of being married reduces the result on the gastrointestinal symptoms domain by an average of 14.961 points in relation to single and widowed status;-The lack of professional activity increases the result on the gastrointestinal symptoms domain by an average of 19.389 points.

The R^2^ coefficient for this model was 18.17%, which means that 18.17% of the variability of the gastrointestinal symptoms domain result was explained by the variables included in the model (see Table 7).

#### 3.4.5. Peripheral Neuropathy Domain

Table 8 presents the results of a multivariate analysis for the QLQ-OV28 scale in the peripheral neuropathy domain. 

A multivariate linear regression model showed that professional inactivity increases the peripheral neuropathy domain score by an average of 20.597 points.

The R^2^ coefficient for this model was 32.57%, which means that 32.57% of the variability of the peripheral neuropathy domain result was explained by the variables included in the model (see Table 8).

#### 3.4.6. Hormonal/Menopausal Symptoms Domain

Table 9 presents the results of a multivariate analysis for the QLQ-OV28 scale in the hormonal/menopausal symptoms domain. The multivariate linear regression model showed the following:-Being married reduces the score on the hormonal/menopausal symptoms domain by an average of 17.222 points in relation to being single;-Widowhood reduces the score on the hormonal/menopausal symptoms domain by an average of 22.994 points in relation to being single.

The R^2^ coefficient for this model was 12.76%, which means that 12.76% of the variability of the hormonal/menopausal symptoms domain result was explained by the variables included in the model (see Table 9).

#### 3.4.7. Other Side Effects of Chemotherapy Domain

Table 10 presents the results of a multivariate analysis for the QLQ-OV28 scale in the other side effects of chemotherapy domain. The multivariate linear regression model showed the following:-A lack of professional activity increases the result on the other side effects of chemotherapy domain by an average of 14.401 points.

The R^2^ coefficient for this model was 22.34%, which means that 22.34% of the other side effects of chemotherapy domain result was determined by the variables included in the model (see Table 10).

#### 3.4.8. Hair Loss Domain

Table 11 presents the results of a multivariate analysis for the QLQ-OV28 scale in the hair loss domain. The model showed that a lack of professional activity increases the result on the hair loss domain by an average of 21.947 points.

The R^2^ coefficient for this model was 24.15%, which means that 24.15% of the variability of the hair loss domain result was explained by the variables included in the model (see Table 11).

## 4. Discussion

Research on quality of life primarily aims to show how the disease, symptoms or treatment affect patients and how it relates to different life domains/activities. Counterfactually, few research papers analyse the quality of life in women with ovarian cancer treated with chemotherapy. The available scientific papers approach the topic in a general way, cover different types of cancer therapies, and often do not assess the impact of clinical and demographic factors on quality of life, nor do they compare the stages of chemotherapy administration. It should also be noted that in the few studies cited, the methodology, the research tools used, the group sizes, and the determinants analysed differ significantly from our research.

### 4.1. Quality of Life in Women with Ovarian Cancer during Chemotherapy

In the present study, Polish ovarian cancer patients undergoing first-line chemotherapy had an increasingly poor quality of life assessment with each study stage (four stages). Cytostatics treatment significantly affected overall health, including worsening symptoms/complications and emotional, cognitive, physical, and social functioning. 

Statistically significant differences in the assessment of quality of life were observed in areas related to the side effects of chemotherapy. Comparing the results of the first stage of the study with the results of the fourth stage, the following symptoms were observed: nausea and vomiting, loss of appetite, peripheral neuropathy, and hair loss.

Campbell et al. obtained similar results [14] when they investigated the impact of chemotherapy on women with recurrent ovarian cancer and the quality of their lives. They also used the EORTC QLQ-C30 and EORTC QLQ-OV28 questionnaires as well as the MOST-T35 (Measure of Ovarian Symptoms and Treatment Concerns). In their analyses, symptoms originating from the gastrointestinal tract related to symptomatic neuropathy, nausea and vomiting, and psychiatric symptoms negatively affected the quality of life. Also, the patients presented a low quality of life in the functioning and general health questionnaires. In a study conducted by Nho et al. [15], it was found that successive cycles of chemotherapy, where the patients received either platinum or taxane (and, above all, the resultant numerous side effects), worsened their quality of life. Mental stress, fatigue, pain, abdominal discomfort, flu-like symptoms, fluid accumulation, and peripheral neuropathy negatively affected the quality of life of ovarian cancer patients undergoing adjuvant chemotherapy, especially in patients with depression and high anxiety levels, as reported by Hwang et al. [16]. Low quality of life, especially in the area of physical and emotional functioning, and susceptibility to mood disorders, were presented by patients in the study by Shirala et al. [17]. Deterioration of quality of life in women with ovarian cancer was also observed by Chase et al. [18], Tachata et al. [19], Słoniewski et al. (platinum-based chemotherapy) [20], Lee et al. [21], Sarkar et al. [22], Śniadecki et al. (intraperitoneal chemotherapy—IPC) [23], Bhugwandass et al. (chemotherapy type: carboplatin–paclitaxel, cisplatin–paclitaxel, cisplatin–etoposide, and cyclophosphamide–carboplatin) [24], and Perkowska et al. [25] as a result of chemotherapy. In a study by Lee et al. [26], only 15% (out of approximately 948) of patients indicated improved quality of life after completion of platinum-based chemotherapy. Perkowska et al. [5] and Smorąg et al. [27] linked poor quality of life to depression in women in their studies. Kozaka [28], in her review article, clearly emphasised that treatment-related side effects, including physical and emotional burdens, significantly affect women’s quality of life. 

Plotti et al. (Carboplatin and Paclitaxel) [29], Kim et al. (hyperthermic intraperitoneal chemotherapy—HIPEC) [30], Blagden et al. (carboplatin and paclitaxel) [31], and Penar-Zadarko et al. [32] obtained different results, indicating that chemotherapy does not affect the quality of life in a way that reduces it or keeps it stable [33]. Pergialiotis et al. [34] indicated that the type of treatment used (in this case, combination therapy with taxanes and platinum) affected quality of life. Also, in Sompolska-Rzechuła [4], women undergoing chemotherapy rated their quality of life positively in the social, functional, emotional, and physical spheres, and adverse symptoms did not affect their well-being negatively.

### 4.2. Selected Parameters and the Quality of Life of Women with Ovarian Cancer during Chemotherapy

The research has shown that age is one of the factors that strongly determines quality of life. Depending on the number of years and the subsequent study stage, the women felt that their quality of life was affected differently by their functioning, symptoms, or sense of their health. For example, women over 60 were not affected by sexual problems, hormonal/menopausal symptoms, or their body perception. In contrast, higher levels of fatigue, pain, shortness of breath, or gastrointestinal symptoms were reported to affect quality of life. Alternatively, regarding the effect of nausea and vomiting on quality of life, at Stage III of the study, this symptom correlated strongly in 51–60-year-old women. However, Stage IV did not significantly affect this age group, indicating that life stage (age) and related functioning, social roles, and expectations are reflected in the perception of quality of life. 

In Nho et al. [15], age was also a factor in reduced quality of life. According to Zhou et al. [35], poor quality of life was also related to age (older age influenced poorer functioning in the physical area, whereas younger age influenced poorer functioning in the mental area). Older people (≥70 years of age), as reported by Walree et al. [36], had a lower sense of the quality of life than younger people (<70 years of age), especially in the domains of general health and in all functional subscales, except for emotional functioning. Panoskaltsis et al. [37] performed a multivariate analysis of the effect of age on the quality of life in women with ovarian cancer undergoing intraperitoneal chemotherapy in hypothermia (HIPEC). Advanced age (>65) was a factor that negatively affected patient survival. In Sompolska-Rzechuła [4], women aged 63–88 had a higher quality of life in terms of social well-being, emotional well-being, functional well-being, and perceived complaints, and women aged 24–63 years had a higher quality of life in terms of physical well-being. In the present study, the severity of the disease had an impact on some quality of life domains, both in the functional areas and the symptoms present, being variable depending on the phase of the study. In our study, the multivariate linear regression model showed that lack of professional activity lowers the quality of life, especially because the other side effects of chemotherapy domain, as well as hair loss in Stage IV of the study.

Chase et al. [18], Friendlander et al. [38], Tachata et al. [19] (disease stage adversely affected the gastrointestinal variable in the patients), and Zhou et al. [35] also observed the impact of the disease stage on quality of life. In contrast, Koole et al. [39], who evaluated the effect of intraperitoneal hyperthermia chemotherapy in Stage III ovarian cancer, showed no negative impact on quality of life. Similarly, Sampolska-Rzechuła et al. [4] found that functional well-being, social well-being, and perceived discomfort were not significantly associated with disease stage. 

The research was conducted on the Polish population and should only be related to this population. The authors recommend performing future studies in other national groups and making comparisons, which could provide interesting results and be the subject of interesting discussion.

A significant limitation of our research is the small size of the group. This may limit the ability to conclusively determine the impact of chemotherapy on quality of life. However, it was only possible to recruit the women who met the established criteria during the study period.

The quality of life of the subjects was not assessed by taking into account the scope of the surgery and histologic tumour type. The researchers knew the results of laboratory tests for the examined women, such as the level of haemoglobin, erythrocytes, leukocytes, neurocytes, platelets, and the CA125 antigen. Using multivariate analysis, the article considered these parameters in assessing the quality of life.

The study did not include a comparison group of women without ovarian cancer or women with ovarian cancer who had not received chemotherapy. This may limit the ability to conclusively determine the impact of chemotherapy on quality of life.

This study was partially conducted during the COVID-19 pandemic; however, to maintain the original assumptions, we did not study the impact of COVID-19, which could potentially affect the results.

The innovative value of this study is that the results are paramount and needed in developing treatment and nursing programs to reduce the side effects of chemotherapy, for example, the prevention of nausea and vomiting, appetite enhancement, and coping with hair loss, especially for older and retired women.

## 5. Conclusions

Deterioration of physical, emotional, cognitive, and social functioning during successive rounds of chemotherapy is associated with the need for improving and intensifying the care associated with cancer treatment.

Gastrointestinal symptoms during treatment require continuous monitoring and evaluation for adjustments to supportive management.

Side effects accompanying treatment for ovarian cancer patients require support from psychologists, dieticians, nurses, and physiotherapists.

The obtained results will help to create a strategy for reducing the severity of chemotherapy side effects, which may improve the quality of life of women with ovarian cancer.

The implications of the above research can be a source of education for patients undergoing chemotherapy, as an indicator of what complications and intensity they can expect at each stage of treatment, additionally taking into account their age. For medical staff (nurses and doctors), quality of life research can support a holistic approach to patients who receive chemotherapy, including preventive action to reduce ailments/complications.

The research was conducted in Poland, only in one city. An interesting prospect would be to extend this study to other centres, including foreign ones, and to conduct comparative studies to see if all patient populations respond the same to chemotherapy using different types of cytostatics. 

## Figures and Tables

**Figure 1 healthcare-11-02596-f001:**
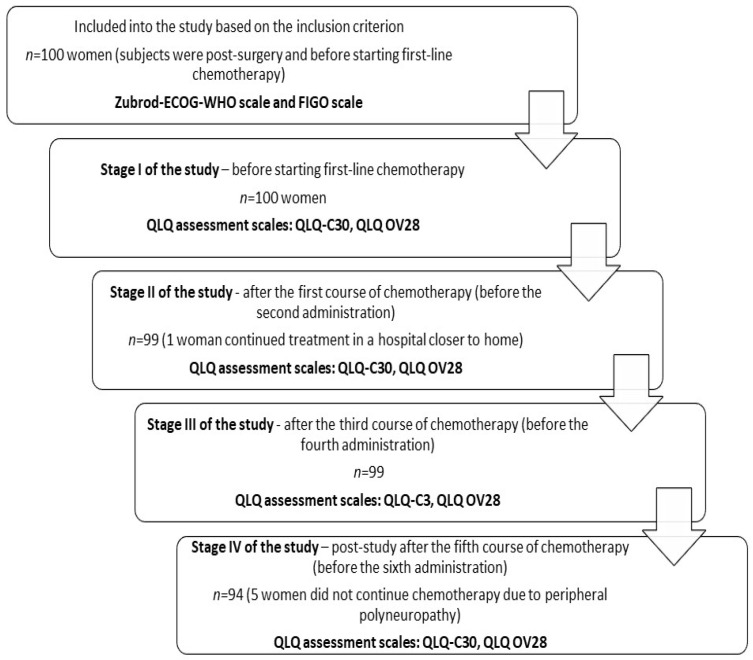
Course of the study.

**Table 1 healthcare-11-02596-t001:** Characteristics of patients upon enrolment in the study.

Variable	Number of Patients (*n*)
Age range (years)	
≤50	14
51–60	21
≥61	65
Marital status	
Single	12
Married	54
Widow	34
Education	
Elementary	13
Vocational	30
Secondary	38
Higher	19
Professional activity	
Professionally active	36
Pension	64
Comorbidities	
YES	59
NO	41
FIGO stage	
II	22
III	68
IV	10

**Table 3 healthcare-11-02596-t003:** Comparison of patients’ quality of life at different study stages based on the QLQ-OV28 functional and symptom scales.

Quality of Life		Stage I	Stage II	Stage III	Stage IV	*p*
Perception of one’s own body	Mean ± SD	30.5 ± 23.81	43.1 ± 24.4	54.71 ± 28.87	58.87 ± 31.84	*p* < 0.001 *
	median	33.33	33.33	66.67	66.67	
	quartiles	0–33.33	33.33–66.67	33.33–66.67	33.33–83.33	IV, III > II > I
Sexuality	Mean ± SD	6.43 ± 11.04	2.34 ± 6.64	1.93 ± 6.25	1.23 ± 5.2	*p* < 0.001 *
	median	0	0	0	0	
	quartiles	0–16.67	0–0	0–0	0–0	I > II, III, IV
Approach to disease/treatment	Mean ± SD	63.67 ± 24.46	73.79 ± 19.65	82.27 ± 17.16	86.53 ± 19.27	*p* < 0.001 *
	median	66.67	77.78	77.78	100	
	quartiles	44.44–77.78	55.56–88.89	66.67–100	77.78–100	IV > III > II > I
Gastrointestinal symptoms	Mean ± SD	39.52 ± 21.5	41.32 ± 19.02	47.14 ± 18.13	51.32 ± 21.15	*p* < 0.001 *
	median	38.1	42.86	42.86	47.62	
	quartiles	28.57–53.57	28.57–52.38	33.33–57.14	34.52–66.67	IV > III > II.I
Peripheral neuropathy	Mean ± SD	19.44 ± 23.39	42.54 ± 24.28	59.15 ± 22.04	71.04 ± 21.57	*p* < 0.001 *
	median	11.11	33.33	66.67	66.67	
	quartiles	0–33.33	33.33–66.67	33.33–66.67	55.56–97.22	IV > III > II > I
Hormonal/menopausal symptoms	Mean ± SD	15.17 ± 20.25	18.18 ± 20.63	21.38 ± 21.57	25.53 ± 24.77	*p* < 0.001 *
	median	0	16.67	16.67	33.33	
	quartiles	0–33.33	0–33.33	0–33.33	0–33.33	IV > III > II.I
Other side effects of chemotherapy	Mean ± SD	26.53 ± 19.21	36.8 ± 17.78	47.68 ± 17.31	53.4 ± 16.92	*p* < 0.001 *
	median	20	40	46.67	53.33	
	quartiles	13.33–40	26.67–46.67	36.66–60	40–66.67	IV > III > II > I
Hair loss	Mean ± SD	8.67 ± 21.25	44.61 ± 22.57	65.96 ± 26.43	70.2 ± 21.33	*p* < 0.001 *
	median	0	33.33	66.67	66.67	
	quartiles	0–0	33.33–66.67	50–83.33	66.67–83.33	III > IV > II.I

*p*—Friedman test + post hoc analysis (Wilcoxon matched pair tests with Bonferroni correction); * statistically significant relationship (*p* < 0.05). QLQ-OV28: functional scales/symptom scales—higher value indicates greater severity of problems/symptoms and lower quality of life; range 0–100.

**Table 4 healthcare-11-02596-t004:** Women’s quality of life according to the QLQ-OV28 in perception of one’s own body domain at Stage IV.

Variable		Regression Coefficient	95%CI		*p*
Age	≤50	ref.			
	51–60	−9.722	−35.549	16.104	0.463
	≥61	−38.395	−66.903	−9.887	0.01 *
Marital status	Single	ref.			
	Married	−10.808	−32.485	10.87	0.331
	Widow	−19.17	−42.553	4.212	0.112
Education	Elementary	ref.			
	Vocational	−8.256	−29.952	13.439	0.458
	Secondary	−4.936	−26.961	17.09	0.662
	Higher	0.866	−26.603	28.335	0.951
Professional activity	Professionally active	ref.			
	Pension	34.2	7.019	61.382	0.016 *
Comorbidities	No	ref.			
	Yes	−1.231	−15.145	12.682	0.863
FIGO stage	II	ref.			
	III	0.808	−14.928	16.544	0.92
	IV	0.598	−25.149	26.345	0.964

*p*—multiple linear regression; * statistically significant (*p* < 0.05).

**Table 5 healthcare-11-02596-t005:** Women’s quality of life according to the QLQ-OV28 in the sexuality domain at Stage IV.

Variable		Regression Coefficient	95%CI		*p*
Age	≤50	ref.			
	51–60	−2.055	−6.219	2.109	0.336
	≥61	−3.944	−8.541	0.652	0.096
Marital status	Single	ref.			
	Married	1.845	−1.651	5.34	0.304
	Widow	1.501	−2.269	5.271	0.437
Education	Elementary	ref.			
	Vocational	−0.055	−3.553	3.443	0.975
	Secondary	−0.719	−4.27	2.833	0.693
	Higher	0.963	−3.466	5.392	0.671
Professional activity	Professionally active	ref.			
	Pension	−0.731	−5.114	3.651	0.744
Comorbidities	No	ref.			
	Yes	0.033	−2.21	2.276	0.977
FIGO stage	II	ref.			
	III	0.17	−2.368	2.707	0.896
	IV	−0.458	−4.609	3.693	0.829

*p*—multiple linear regression.

**Table 6 healthcare-11-02596-t006:** Women’s quality of life according to the QLQ-OV28 in the approach to disease/treatment domain at Stage IV.

Variable		Regression Coefficient	95%CI		*p*
Age	≤50	ref.			
	51–60	2.754	−13.096	18.604	0.734
	≥61	−4.334	−21.83	13.162	0.629
Marital status	Single	ref.			
	Married	−4.036	−17.339	9.268	0.554
	Widow	0.651	−13.699	15.001	0.929
Education	Elementary	ref.			
	Vocational	−9.135	−22.45	4.179	0.182
	Secondary	−2.042	−15.559	11.476	0.768
	Higher	−0.968	−17.826	15.89	0.911
Professional activity	Professionally active	ref.			
	Pension	14.151	−2.531	30.832	0.1
Comorbidities	No	ref.			
	Yes	−4.029	−12.568	4.509	0.358
FIGO stage	II	ref.			
	III	3.802	−5.855	13.459	0.443
	IV	3.993	−11.809	19.794	0.622

*p*—multiple linear regression.

**Table 7 healthcare-11-02596-t007:** Women’s quality of life according to the QLQ-OV28 in the gastrointestinal symptoms domain at Stage IV.

Variable		Regression Coefficient	95%CI		*p*
Age	≤50	ref.			
	51–60	4.447	−12.25	21.143	0.603
	≥61	−3.976	−22.406	14.454	0.674
Marital status	Single	ref.			
	Married	−14.961	−28.976	−0.947	0.039 *
	Widow	−14.078	−29.194	1.039	0.072
Education	Elementary	ref.			
	Vocational	−9.727	−23.752	4.299	0.178
	Secondary	−5.463	−19.702	8.776	0.454
	Higher	−3.348	−21.106	14.41	0.713
Professional activity	Professionally active	ref.			
	Pension	19.389	1.816	36.961	0.033 *
Comorbidities	No	ref.			
	Yes	−1.886	−10.881	7.109	0.682
FIGO stage	II	ref.			
	III	0.476	−9.697	10.65	0.927
	IV	5.285	−11.361	21.93	0.535

*p*—multiple linear regression; * statistically significant (*p* < 0.05).

**Table 8 healthcare-11-02596-t008:** Women’s quality of life according to the QLQ-OV28 in the peripheral neuropathy domain at Stage IV.

Variable		Regression Coefficient	95%CI		*p*
Age	≤50	ref.			
	51–60	12.852	−2.607	28.31	0.107
	≥61	10.614	−6.45	27.677	0.226
Marital status	Single	ref.			
	Married	−5.969	−18.944	7.006	0.37
	Widow	−5.302	−19.297	8.694	0.46
Education	Elementary	ref.			
	Vocational	−1.427	−14.412	11.559	0.83
	Secondary	0.828	−12.355	14.011	0.902
	Higher	−3.805	−20.246	12.637	0.651
Professional activity	Professionally active	ref.			
	Pension	20.597	4.328	36.867	0.015 *
Comorbidities	No	ref.			
	Yes	−7.103	−15.431	1.224	0.098
FIGO stage	II	ref.			
	III	0.533	−8.886	9.952	0.912
	IV	3.236	−12.175	18.646	0.682

*p*—multiple linear regression; * statistically significant (*p* < 0.05).

**Table 9 healthcare-11-02596-t009:** Women’s quality of life according to the QLQ-OV28 in the hormonal/menopausal symptoms domain at Stage IV.

Variable		Regression Coefficient	95%CI		*p*
Age	≤50	ref.			
	51–60	−8.082	−28.274	12.109	0.435
	≥61	−20.106	−42.394	2.182	0.081
Marital status	Single	ref.			
	Married	−17.222	−34.169	−0.274	0.05 *
	Widow	−22.994	−41.275	−4.713	0.016 *
Education	Elementary	ref.			
	Vocational	6.841	−10.121	23.803	0.432
	Secondary	1.493	−15.726	18.713	0.865
	Higher	2.829	−18.647	24.305	0.797
Professional activity	Professionally active	ref.			
	Pension	10.269	−10.982	31.52	0.346
Comorbidities	No	ref.			
	Yes	−0.77	−11.648	10.108	0.89
FIGO stage	II	ref.			
	III	−0.009	−12.311	12.294	0.999
	IV	−0.873	−21.003	19.256	0.932

*p*—multiple linear regression; * statistically significant (*p* < 0.05).

**Table 10 healthcare-11-02596-t010:** Women’s quality of life according to the QLQ-OV28 in the other side effects of chemotherapy domain at Stage IV.

Variable		Regression Coefficient	95%CI		*p*
Age	≤50	ref.			
	51–60	11.071	−1.941	24.084	0.099
	≥61	4.65	−9.714	19.014	0.528
Marital status	Single	ref.			
	Married	−3.607	−14.529	7.315	0.519
	Widow	−0.543	−12.324	11.238	0.928
Education	Elementary	ref.			
	Vocational	−2.964	−13.895	7.967	0.597
	Secondary	−2.055	−13.152	9.043	0.718
	Higher	−3.076	−16.916	10.764	0.664
Professional activity	Professionally active	ref.			
	Pension	14.401	0.706	28.097	0.042 *
Comorbidities	No	ref.			
	Yes	0.25	−6.76	7.26	0.944
FIGO stage	II	ref.			
	III	0.794	−7.134	8.723	0.845
	IV	0.6	−12.373	13.572	0.928

*p*—multiple linear regression; * statistically significant (*p* < 0.05).

**Table 11 healthcare-11-02596-t011:** Women’s quality of life according to the QLQ-OV28 in the hair loss domain at Stage IV.

Variable		Regression Coefficient	95%CI		*p*
Age	≤50	ref.			
	51–60	−10.947	−31.034	9.141	0.289
	≥61	−16.865	−39.038	5.309	0.14
Marital status	Single	ref.			
	Married	13.815	−3.045	30.675	0.112
	Widow	3.774	−14.412	21.96	0.685
Education	Elementary	ref.			
	Vocational	−14.839	−31.713	2.035	0.089
	Secondary	−1.959	−19.09	15.172	0.823
	Higher	8.203	−13.162	29.568	0.454
Professional activity	Professionally active	ref.			
	Pension	21.947	0.806	43.088	0.045 *
Comorbidities	No	ref.			
	Yes	−10.49	−21.312	0.332	0.061
FIGO stage	II	ref.			
	III	10.296	−1.943	22.535	0.103
	IV	−7.532	−27.558	12.494	0.463

*p*—multiple linear regression; * statistically significant (*p* < 0.05).

## Data Availability

Data are available from the authors upon reasonable request.

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
