# Peer review of "Quality of Life for Polish Women with Ovarian Cancer during First-Line Chemotherapy"

_healthcare, 2023, doi:10.3390/healthcare11182596_

Round 1

Reviewer 1 Report

Overall, the study provides valuable insights into the quality of life of patients with ovarian cancer. However, I have several concerns about the study that should be addressed before it can be considered for publication.

 1.   The sample size of the study is relatively small (n=100), which may limit the generalizability of the findings. The authors should consider expanding the sample size or acknowledging the limitations of the study due to the small sample size. For example, the authors could discuss the statistical power of their analyses and whether the sample size was sufficient to detect meaningful differences in quality of life between different age groups.

2.   the study was conducted in Poland, and it is unclear whether the results can be generalized to other populations. The authors should discuss the generalizability of their findings and consider conducting similar studies in other countries to confirm their results. For example, the authors could discuss whether the cultural, geographical, and social context of Poland may have influenced the results and whether similar studies have been conducted in other countries.

3.       the study makes several assumptions about the patients' treatment and experiences that may not be accurate or appropriate. The authors should be more transparent about these assumptions and consider conducting additional analyses to confirm their validity. For example, the authors assume that all patients received the same treatment, but it is unclear whether this is the case. The authors could consider conducting sensitivity analyses to assess the impact of different treatment regimens on quality of life.

4.       lack of comparison group: The study did not include a comparison group of women without ovarian cancer or women with ovarian cancer who were not undergoing chemotherapy. This limits the ability to draw conclusions about the impact of chemotherapy on quality of life. I recommend that the authors discuss the limitations of the study in the discussion section and suggest directions for future research.

5.       while the study provides important information about the quality of life of patients with ovarian cancer, it is not clear how the results of the study can be used to improve patient care or inform clinical decision-making

Minor edit is needed

Author Response

Thank you very much for your reviews. At the suggestion of the reviewers, we made we have made changes to make the article more to be more interesting and bring new knowledge to the reader. The English language has been improved too. We tried to adapt to all comments, and changes are marked in red.

Comments and Suggestions for Authors

Overall, the study provides valuable insights into the quality of life of patients with ovarian cancer. However, I have several concerns about the study that should be addressed before it can be considered for publication.

  1. The sample size of the study is relatively small (n=100), which may limit the generalizability of the findings. The authors should consider expanding the sample size or acknowledging the limitations of the study due to the small sample size. For example, the authors could discuss the statistical power of their analyses and whether the sample size was sufficient to detect meaningful differences in quality of life between different age groups.

In the limitations of the study, we will write:

An important limitation of our research is the small size of the group. This may limit the ability to conclude the impact of chemotherapy on quality of life.  However, during the study period, it was possible to recruit only the number of women who met the established criteria.

  1.  the study was conducted in Poland, and it is unclear whether the results can be generalized to other populations. The authors should discuss the generalizability of their findings and consider conducting similar studies in other countries to confirm their results. For example, the authors could discuss whether the cultural, geographical, and social context of Poland may have influenced the results and whether similar studies have been conducted in other countries.

We will write: The research was conducted on the Polish population and should only be related to this population. The authors recommend performing future studies in other national groups and making comparisons, which could provide interesting results and be the subject of interesting discussion.

  1. the study makes several assumptions about the patients' treatment and experiences that may not be accurate or appropriate. The authors should be more transparent about these assumptions and consider conducting additional analyses to confirm their validity. For example, the authors assume that all patients received the same treatment, but it is unclear whether this is the case. The authors could consider conducting sensitivity analyses to assess the impact of different treatment regimens on quality of life.

We relied on information from doctors and we have no reason to question their opinion. The inclusion criterion was based on including patients as similar as possible to each other, therefore we excluded other treatment regimens. The research was conducted reliably. Dr Anna Pleszewa, as a nursing ward of the chemotherapy ward in Wielkopolska Oncology Centre Poznan, supervised the accuracy of these tests and personally collected data from patients.

  1. lack of comparison group: The study did not include a comparison group of women without ovarian cancer or women with ovarian cancer who were not undergoing chemotherapy. This limits the ability to draw conclusions about the impact of chemotherapy on quality of life. I recommend that the authors discuss the limitations of the study in the discussion section and suggest directions for future research.

In the limitations of the study, we will write: The study did not include a comparison group of women without ovarian cancer or women with ovarian cancer who had not received chemotherapy. This may limit the ability to conclude the impact of chemotherapy on quality of life.

  1. while the study provides important information about the quality of life of patients with ovarian cancer, it is not clear how the results of the study can be used to improve patient care or inform clinical decision-making

The conclusions were supplemented with the following description

The implications of the above research can be a source of education for patients undergoing chemotherapy, as an indicator of what complications and what intensity they can expect at each stage of treatment, additionally taking into account their age. For medical staff (nurses and doctors), quality of life research is a perspective for a holistic approach to patients who receive chemotherapy, taking preventive action in advance to reduce ailments/complications. 

Reviewer 2 Report

The authors have drafted a manuscript called “Quality of life in Polish women with ovarian cancer during 2 first-line chemotherapy Concerning the age of the patients”. They proved that women with ovarian cancer undergoing chemo need additional

support. A few corrections are necessary before publishing.

1. The layout of Table 1 is very confusing. A better design is necessary. Also, Race/ethnicity, Family history of ovarian and or breast cancer, State at diagnosis, and Histology should be included.

2. Other chemo reagents should be included or at least mentioned in the discussion. Do different reagents impact the conclusions?

Moderate editing of English language required

Author Response

Thank you very much for your reviews. At the suggestion of the reviewers, we made we have made changes to make the article more to be more interesting and bring new knowledge to the reader. The English language has been improved too. We tried to adapt to all comments, and changes are marked in red.

Comments and Suggestions for Authors

The authors have drafted a manuscript called “Quality of life in Polish women with ovarian cancer during 2 first-line chemotherapy Concerning the age of the patients”. They proved that women with ovarian cancer undergoing chemo need additional support. A few corrections are necessary before publishing.

  1. The layout of Table 1 is very confusing. A better design is necessary. Also, Race/ethnicity, Family history of ovarian and or breast cancer, State at diagnosis, and Histology should be included.

Table 1 has been corrected white race, and Polish nationality - were the selection criteria for the study. Family history of ovarian and or breast cancer, State at diagnosis, and Histology – these variables were not included in the studies. The authors thank you very much for this comment, these variables will be included in the new project in cooperation with the project in cooperation with Higher School of Nursing in Lisbon.

  1. Other chemo reagents should be included or at least mentioned in the discussion. Do different reagents impact the conclusions?

Supplemented

Reviewer 3 Report

1. Define QLQ-C30, and QLQ OV28.

2. Figure 1 is a text not a Figure. Which part of the text included in Figure 1.

3. List of abbreviation should be included in the manuscript for better understanding to readers

4. The study ran from May 1, 2019 to October 30, 2020. This period includes COVID-19 phase also. What are the impacts of COVID-19 in the present studies.

5. Table 1 is not clear. What the authors want to present? 

6. Add footnote in the table

7. Explain the legend caption in the Figure.

8. Table 7, somewhere authors used "." and somewhere ",". why?

9. Elaborate the outcomes and future perspectives of the current investigation.

Minor editing of English language required

Author Response

Thank you very much for your reviews. At the suggestion of the reviewers, we made we have made changes to make the article more to be more interesting and bring new knowledge to the reader. The English language has been improved too. We tried to adapt to all comments, and changes are marked in red.

  1. Define QLQ-C30, and QLQ OV28.

The full name of the scales is given. In the method section the scales are described in detail

  1. Figure 1 is a text not a Figure. Which part of the text included in Figure 1.

Figure 1 has been removed and the treatment regimen is presented as text

  1. List of abbreviation should be included in the manuscript for better understanding to readers

Abbreviations in the text and tables have been removed for clarity and better understanding for the reader, replacing them with the full name

  1. The study ran from May 1, 2019 to October 30, 2020. This period includes COVID-19 phase also. What are the impacts of COVID-19 in the present studies.

Part of the study occurred during the COVID-19 pandemic, but we couldn't have predicted it when we started the project. To maintain the original assumptions, we did not consider the pandemic's impact. However, we are aware that the pandemic may have potentially affected the results. Therefore, in the limitations of the study, we will write: This study was partially conducted during the COVID-19 pandemic; however, to maintain the original assumptions, we did not study the impact of COVID-19, which could potentially affect results.

  1. Table 1 is not clear. What the authors want to present? 

Table 1 has been corrected

  1. Add footnote in the table

Supplemented

  1. Explain the legend caption in the Figure.

Figure 1 has been removed and the treatment regimen is presented as text

  1. Table 7, somewhere authors used "." and somewhere ",". why?

Corrected

  1. Elaborate the outcomes and future perspectives of the current investigation.

Discussion of the results has been supplemented. The new project involves conducting a study among the same women to compare the quality of life several years after the end of chemotherapy

Round 2

Reviewer 1 Report

Thanks the authors for considering the comments.

To my comment 3, the authors may misunderstood my point. To me it is unclear if all the patients received the same treatment. This is important because side effects of different treatment may be different which may confound the analysis and lead to wrong conclusions. I am not questioning the treatment decisions. In addition, being a nurse ward does not necessarily mean the statistical analysis supervised by her is appropriate because this is typically outside of the professional area of a nurse ward.

NA

Author Response

Thank you very much for your review

To my comment 3, the authors may misunderstood my point. To me it is unclear if all the patients received the same treatment. This is important because side effects of different treatment may be different which may confound the analysis and lead to wrong conclusions. I am not questioning the treatment decisions. In addition, being a nurse ward does not necessarily mean the statistical analysis supervised by her is appropriate because this is typically outside of the professional area of a nurse ward.
A. All women received the same treatment regimen, which was the main criterion for selecting women for the study. Therefore, in the first-line chemotherapy treatment manuscript, we added this criterion:
The following inclusion criteria were adopted for the study:
a)        post-operative patients scheduled for first-line chemotherapy, and first-line chemotherapy treatment

Reviewer 3 Report

The authors didn't understand my comment no. 3. I just recommend to add a separate section of list of abbreviation.

Author Response

Thank you very much for your review.
The authors didn't understand my comment no. 3. I just recommend to add a separate section of list of abbreviation.

We added a list of abbreviations and it was placed before the references.